# Investigation of the Evolution of Schmid Factors (SF) in 316 Stainless Steel during In Situ Plastic Deformation

Xiaofeng Wan [1], Jin Wang [1], Penghou Li [2], Jianguo Chen [3,4,*] and Xiao Wang [1,*]

1  School of Mechanical Engineering, Nantong University, Nantong 226019, China
2  Resources Power Xiantao Co., Ltd., Xiantao 433000, China
3  Institute of Public Safety Research, Tsinghua University, Beijing 100084, China
4  Hefei Institute for Public Safety Research, Tsinghua University, Hefei 230601, China
*  Correspondence: chenjianguo@tsinghua.edu.cn (J.C.); wxm1273@sina.com (X.W.)

**Abstract:** The Schmid factor (SF) is a critical parameter in crystal plasticity research that is often used to evaluate the level of difficulty in activating the slip systems within a grain. The evolution process and change mechanism of SF in 316 austenitic stainless steel during plastic deformation were investigated in this paper by using the in situ electron backscatter diffraction (EBSD) technique. The results showed that the average Schmid factor of global grains was highest in the original state, but after stretching, multiple rotation paths appeared in the grain, and the SF presented a monotonically decreased tendency with the increase in plastic strain degree. Numerical computation revealed that the decrease of SF was mainly governed by the change in $\varphi$ angle, i.e., the angle between loading direction and slip plane normal increased inside the grains after the lattice rotation, which caused the slip plane to move parallel to the loading direction. The higher $\varphi$, the lower its cosine, which corresponds to low shear stress acting on the slip plane and could increase the difficulty of crystal slip.

**Keywords:** 316 steel; Schmid factor; change mechanism; lattice rotation

## 1. Introduction

Type 316 austenitic stainless steel (ASS), next only to 304 ASS, is the second most widely used model in the world for austenitic stainless steel [1,2]. Generally, 316 ASS has excellent fabricability, heat resistance, ductility, weldability, and good resistance to stress corrosion, which are extensively employed in the architectural industry, automotive industry, mechanical industry, and energy source industry [3–5]. Nevertheless, these 316 ASS steel structural components utilized in engineering practices often suffer from tension loads or high energy impacts, which may cause severe plastic deformation during their service life [6].

In general, with the continuous plastic deformation of metals under external loading, the plastic damage will accumulate irreversibly and lead to the deterioration of the mechanical properties of the material, such as a reduction in strength and stiffness. The plastic deformation of polycrystalline materials is mainly manifested as the deformation of each grain with a different slip mode, and the slip system can only be started when the shear stress on the slip surface reaches a specific value. The Schmid factor (SF) is used as a parameter to evaluate the level of ease to activate dislocation movement within a grain [7]; therefore, it is necessary to study the change in SF of 316 ASS steel during plastic tensile in order to explore its deformation mechanism.

Currently, studies are being conducted on 316 ASS based on the in situ electron backscatter diffraction (EBSD) technique; however, most of them focus on the measurement of crystal orientation [8,9], not on the change of SF during the plastic deformation. Considering the wide employment of these structural steel components, more research attention should be devoted to this area. In this research, the 316 ASS was selected as the

research object for in situ EBSD uniaxial tensile tests to explore the change rule of SF and its internal cause.

## 2. Experiments

The composition of 316 ASS used here was: C: 0.04, Cr: 16.38, Ni: 9.73, Mo: 2.15, Mn: 0.932, Si:0.492, Cu: 0.275, Co: 0.351, and Fe: Bal (wt. %). The specimen was fabricated into a dog-bone shape with nominal dimensions of 40 mm (length) × 10 mm (width) × 0.8 mm (thickness). The initial state of the 316 ASS was the hot-rolled sheet, and the tensile specimen was fabricated from this sheet (see Figure 1a). To perform the in situ tensile experiment, a Zeiss-Sigma 300 type Scanning Electron Microscope (Carl Zeiss, Oberkochen, Germany) equipped with an Oxford-SYMMETRY EBSD detector (Oxford Instruments, Abingdon, UK) as well as a Kammrath & Weiss type tensile stage (Kammrath & Weiss GmbH, Schwerte, Germany) (see Figure 1b) were used in combination. The tensile rate of the 316 ASS specimen was 10 μm/s, and the strain of the fractured specimen was 68.76%. To produce varying degrees of plastic deformation, the 316 ASS was stretched to several different degrees of strain (see Figure 1c). The EBSD scanning was carried out on the external surface, at the center of the tensile specimen. The specimen coordinates were given as $X_0$, $Y_0$, and $Z_0$, which were parallel to the coordinate systems of the tensile stages $X_1$, $Y_1$, and $Z_1$, respectively (see Figure 1a,b). The EBSD measurement was conducted at a spatial resolution of 1 μm, and the results were analyzed using the ATEX-4.03 software [10,11].

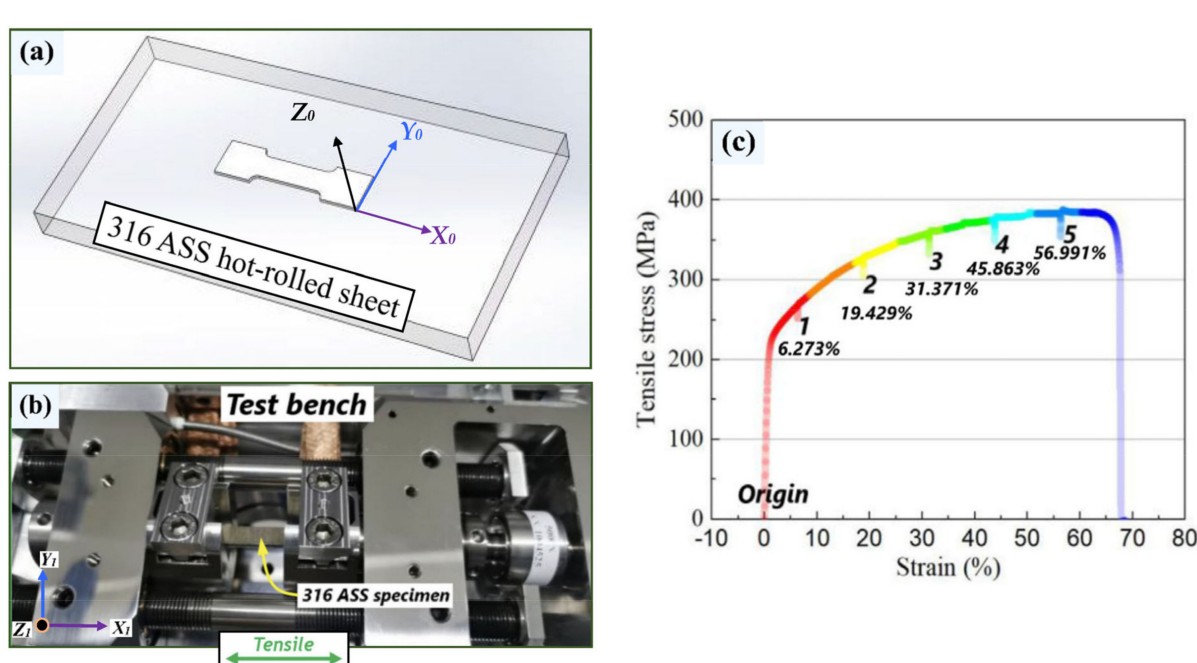

**Figure 1.** In situ tensile test of the 316 ASS. (**a**) The specimen coordinates diagram; (**b**) the test stage; (**c**) the stress–strain curve.

## 3. Results and Discussions

Figure 2a shows the grain orientation (GO) maps of 316 ASS specimens at several plastic strain conditions in a similar viewing area. In the origin specimen, the dots in the Inverse Pole Figure (IPF) map represent a distinct, concentrated distribution. For instance, within the mark of the red-dotted circle, the distance between each dot is small, and these dots are found principally for Grain 1 and Grain 2. By contrast, the orientation dots in the IPF map of 316 ASS show much dispersion after tensile point 2, and a number of new orientations emerged, revealing that the lattice rotation occurred after the plastic deformation. Chen and Mao et al. [7] reported that the lattice rotation behavior in an Al–Mg–Si alloy was related to the activation of slip systems and the Schmid factors (SFs).

Zhang and Lu et al. [12] suggested that the orientation of softer grains with large SF slips more easily. Here, the change of SF of 316 ASS during tensile was shown in Figure 2c; the SF was computed using the {111}<110> slip systems of the FCC lattice, and its variation rules were investigated in the way of global grain statistics and individual grain analysis, respectively. Figure 2d first shows the average value of SF for all pixels within the scanning area. The average SF was highest at the original state (0.43718) but decreased monotonically with the increase in deformation degree, even at the end of the plastic stage. The reduction in average SF also indicates that the lattice slip of 316 ASS becomes more difficult with increasing plastic strain. The relative frequency histogram of SF distribution clearly shows that after the plastic tensile, the proportion of SF in the range of 0.40–0.5 decreased and transformed into the range of 0.30–0.4. Generally, the SF can be the parameter to evaluate the ease of slip system activation since a slip system with a larger SF could be easier to move due to the higher magnitude of shear stress. Thus, further studies on the internal relations between lattice orientation, rotation behavior, and SF variation at the level of individual grains are needed.

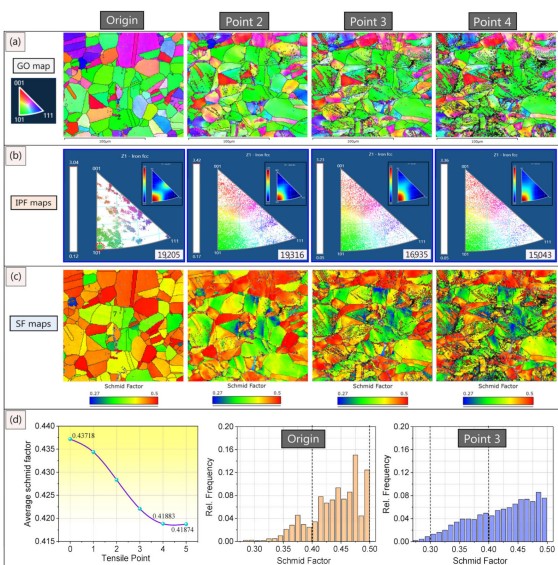

**Figure 2.** The SF change of 316 ASS during the plastic deformation. (**a**) The Grain Orientation map; (**b**) the Inverse Pole Figure (IPF) map; (**c**) the Schmid factors (SF) map; (**d**) the average SF variation and the relative frequency distribution maps of the Schmid factors.

Figure 3a shows the GO map and SF map of a subset region inside the origin specimen, which contains the grains from ID-3 to ID-8. This region is significant in that Grains 3–5 have larger SF values, which stand in sharp contrast to other grains since the SF levels of Grains 6–8 are relatively low. The average lattice orientation (LO) maps of Grains 3, 5, 7, and 8 are shown in Figure 3b, and their specific Euler angle values are also given here. The Miller indices of Grain 3 are (9, 30, 95) [−6.86, −95.01, 30.43], and its activated slip system is (1–11)[110]. Numerical computation showed that the angle $\varphi$ between loading direction $X_1$ and the slip plane normal was 46.79°, and the angle $\lambda$ between $X_1$ and slip direction was approximately 43.92°; thus, Grain 3 has a SF value as high as 0.492. For Grain 5, the $\varphi$ and $\lambda$ were 41.17° and 49.24°, and its SF value was 0.491. In contrast to that, it was found that the $\varphi$, $\lambda$ of Grain 7 and Grain 8 were 57.71°, 41.95° and 63.07°, 32.84°, and the SF values of them were 0.397 and 0.378, respectively. Compared with the large SF of Grain 3 and 5, the high $\varphi$ angle of Grain 7 and 8 was directly responsible for their low SF values. This is most marked in Grain 8, though it has a small angle $\lambda$; the high $\varphi$ (63.07°) results in a low cosine of 0.4529, which significantly lowers the SF value.

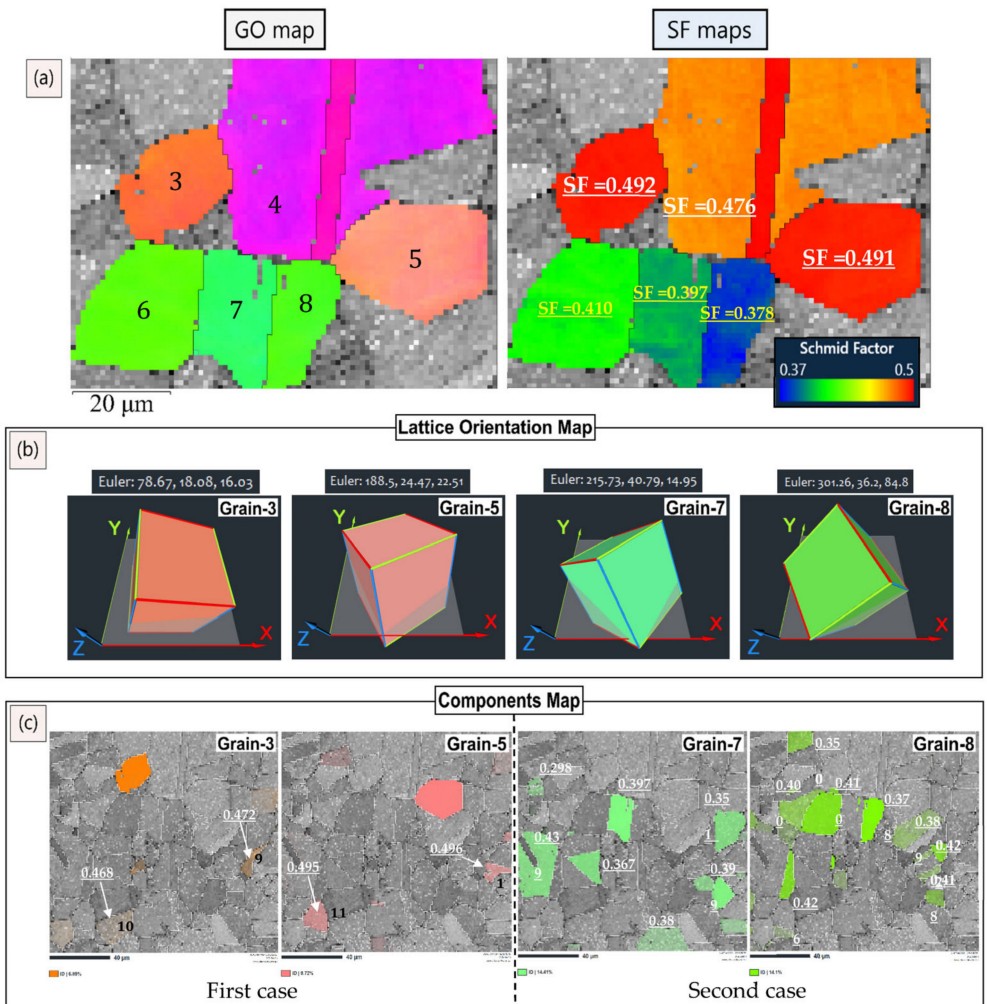

**Figure 3.** The SF of partial grains (Grains 3, 4, 5, 6, 7, 8) of 316 ASS at the origin state. (**a**) The Grain Orientation map and Schmid factors map of partial grains; (**b**) the lattice orientation of Grains 3, 5, 7, and 8; (**c**) the component map of those grains that have <20° misorientation with Grains 3, 5, 7, and 8.

Figure 3c shows the component map of those grains that have an approximate orientation (<20° misorientation angle) with Grains 3, 5, 7, and 8, respectively. In the first case, Grains 9 and Grain 10 have misorientations of nearly 8.11° and 18.3° with Grain 3, and their SF values are 0.472 and 0.468. Grain 11 and Grain 12 have misorientations of 8.51° and 9.89° with Grain 5, and their SF values could go as high as 0.495 and 0.496. In the second case, the SF values of those grains that have the approximate orientation with Grain 7 and Grain 8 are shown in the components map, and it can be seen that the SF values of these grains are significantly smaller. Moreover, it can be seen that the grain number acquired in the second case is much higher than that of the first case. From a consistency perspective, the results indicate that those grains with low SF values may have a similar orientation, while those with high SF values may have an arbitrary orientation instead of a homogeneous orientation.

Figure 4 shows the evolution of the IPF map, the GO map, the LO map, the IPF single pixel (IPF-S) map, and the SF map of Grain 7 during the in situ tensile. Grain 7 is a representative grain since it took the lead in the SF reduction (see Figure 2c). The redistribution of lattice orientation of Grain 7 can be clearly observed in the IPF map as the specimen was stretched to Point 2; multiple rotation paths appeared in this grain, and the maximum rotation angles of paths I, II, III, and IV were approximately 6.48°, 6.08°, 8.34°, and 8.06°, respectively. The occurrence of a new slip system in Grain 7 during tensile may be responsible for this multidirectional rotation since the main slip system in Grain 7 at the

origin state was (1–1–1)[101], but (1–1–1)[01–1], (111)[01–1], and (1–11)[110] appeared at the Point 2 state. To figure out the specific rotation action, individual pixels (A~H) were selected for further analysis.

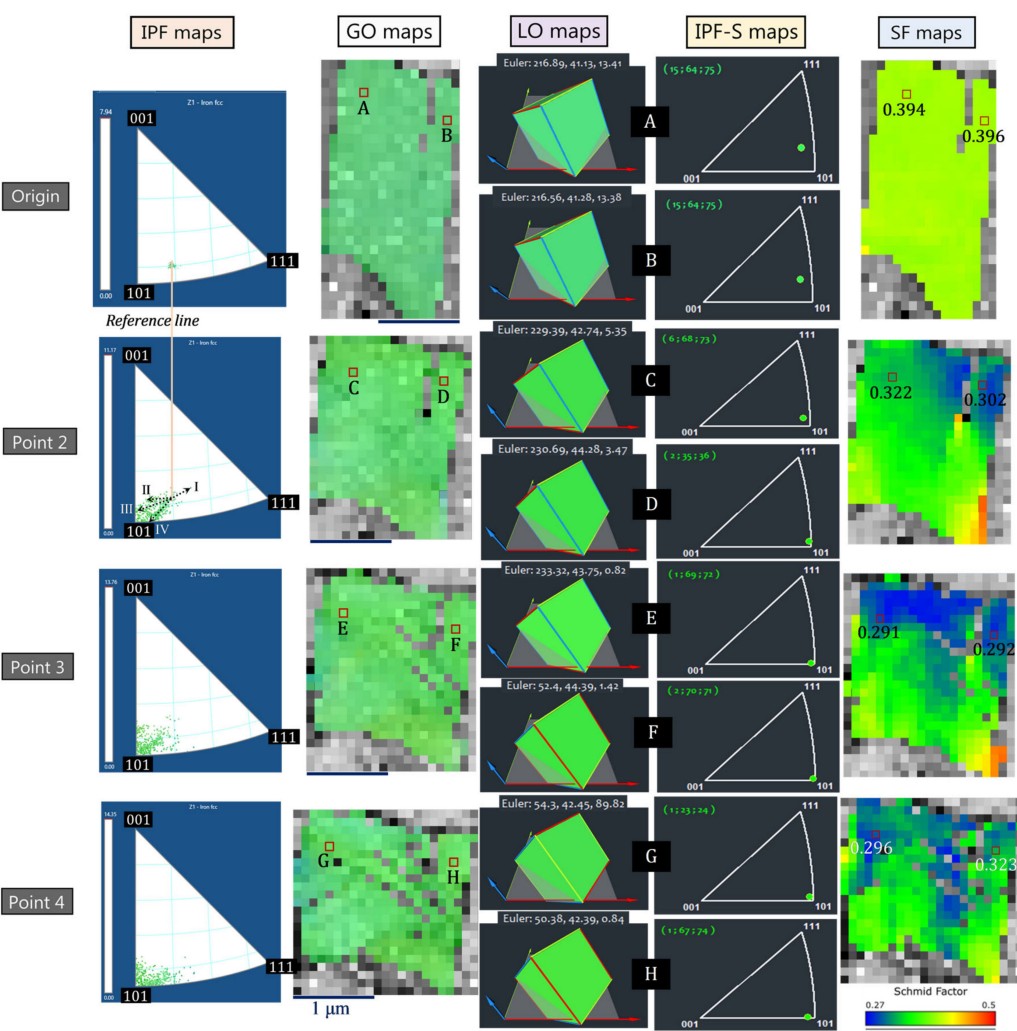

**Figure 4.** The evolution of the IPF map, the GO map, the LO map, the IPF single pixel (IPF-S) map of individual pixels (A~H), and the SF map of Grain 7 during the in situ tensile.

The Miller indices of Pixel A and Pixel B are approximately (15, 64, 75) [−67.31, 62.53, −39.48] and (15, 64, 75)[−67.50, 62.59, −39.07], and their activated slip system is (1–1–1)[101]. The calculated results revealed that the $\varphi$ of Pixel A and Pixel B were approximately 58.55° and 58.30°, and their $\lambda$ values were approximately 40.96° and 41.10°; thus, the SF of these two pixels were 0.394 and 0.396, respectively. After being stretched to Point 2, from Pixel C and Pixel D, it can be seen that the lattice rotation occurred as a result of plastic deformation, which can be observed in the LO map in comparison with the orientation of A-C and B-D pixels. The Miller indices of Pixel C and Pixel D are approximately (6, 68, 73) [−59.61, 61.58, −51.52] and (4.23, 69.69, 71.59) [−59.88, 59.13, −54.02]; the rotation path seems to conform to path IV (see IPF and IPF-S map); and the misorientations between A-C and B-D were approximately 5.62° and 7.63°. Since (1–1–1)[01–1] is one of the main slip systems of Grain 7 in the Point 2 state, the computed results showed that the φ values of Pixel C and Pixel D were approximately 66.28° and 67.96°, and their $\lambda$ values were approximately 36.89° and 36.86°; hence, the SF of Pixel C and Pixel D were 0.322 and 0.302. Compared with Pixel A and Pixel B, the SF of Pixel C and Pixel D were relatively small, and it can be seen that the reduction in SF is principally because of the increase in $\varphi$, i.e., $\varphi_A(58.55°) < \varphi_C(66.28°)$ and $\varphi_B(58.30°) < \varphi_D(67.96°)$. Meanwhile, the higher $\varphi$ also

means that the slip plane tends to be parallel to the loading direction. The higher $\varphi$, the lower its cosine, which results in a low shear stress acting on the slip plane; thus, the slip of the crystal plane became difficult.

After stretching to Point 3, the SF value of some pixels continued to slow down, and the SF of Pixel E and Pixel F were 0.291 and 0.292, and their activated slip systems were both (1–1–1)[101]. The decrease in SF was still due to the increase of $\varphi$ since the calculated results showed that the $\varphi_E$ and $\varphi_F$ were approximately 68.95° and 68.92°, which are larger than $\varphi_C$ (66.28°) and $\varphi_D$ (67.96°). Nevertheless, after stretching to Point 4, it was found that the SF of many pixels increased instead of continuing to decline, as the SF of Pixel G and Pixel H were 0.297 and 0.323, respectively. The Miller indices of Pixel G and Pixel H were approximately (67.5, 0.21, 73.78) [−59.74, −58.53, 54.82] and (0.99, 67.41, 73.86) [62.93, −57.82, 51.93], and the total slip system of Grain 7 was (1–1–1)[101], (1–1–1)[01–1], (1–1–1)[110], (111)[01–1], (111)[10–1], (1–11)[10–1], (1–11)[110]. The computed results showed that, for Pixel G, the max SF occurred on the (111)[10–1] slip system since the $\varphi$ and $\lambda$ values in this system were approximately 68.51° and 35.9°; thus, the SF of this system was approximately 0.296. For Pixel H, the max SF occurred on the (1–1–1)[101] slip system; the $\varphi$ and $\lambda$ values in this system were 66.59° and 35.9°; thus, the SF of this system was approximately 0.323.

Based on the above results and the comparison of Pixel E and Pixel G, it can be deduced that when the SF is small (i.e., 0.291 of Pixel E) and the slip in the (1–1–1)[101] system becomes harder, other systems might activate (i.e., (111)[10–1]) and the slip behavior will translate to the other system, which results in the increase in SF. From a comparison of Pixel F and Pixel H, it can be seen that the activated slip system of each is (1–1–1)[101]; however, the SF increases from Point 3 to Point 4 (i.e., 0.292→0.323) rather than continuing to decrease. The cause of this increase resultingfrom the rotation of local lattice, the $\varphi$ decrease from 68.92° to 66.59°, which lead to the increase of SF.

## 4. Conclusions

In this research, the evolution of the Schmid factor in 316 ASS during uniaxial tensile tests was investigated using global grain statistics and individual grain analysis. The average SF of global grains was highest in the original state but decreased monotonically, which shows that the slip became harder with increasing plastic deformation. Meanwhile, in the origin specimen, some grains had a large SF value while others were relatively low. The main cause of this difference was the difference in $\varphi$ angle. After stretching, multiple rotation paths appeared in the grain, and some new slip systems occurred. The SF value of some pixels showed a decreasing trend, which also resulted from the increase in $\varphi$. As $\varphi$ increased, its cosine became lower, which led to low shear stress acting on the slip plane; thus, the slip of the crystal plane became difficult.

**Author Contributions:** X.W. (Xiaofeng Wan): project administration, supervision. J.W.: formal analysis. P.L.: auxiliary support. J.C.: supervision, technical support. X.W. (Xiao Wang): writing, conception. All authors have read and agreed to the published version of the manuscript.

**Funding:** The Foundation Science Research Program, Nantong city, Grant number JC12022043; the Natural Science Foundation of the Jiangsu Higher Education Institutions of China, Grant number 23KJB460025.

**Data Availability Statement:** The data of these findings can be obtained by contacting the corresponding author.

**Conflicts of Interest:** The authors declare no conflict of interest.

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
