# Peer review of "Investigation of the Evolution of Schmid Factors (SF) in 316 Stainless Steel during In Situ Plastic Deformation"

_crystals, doi:10.3390/cryst13101510_

Round 1

Reviewer 1 Report

The authors used in-situ EBSD technique to study changes in the Schmid factor of 316 stainless steel during plastic deformation. The methodology and results are very well established. The use of 316 stainless steel is detailed and justified and the discussion clarifies the importance of this work to the field. Before publication, I would recommend only a few amends:

  1. “Currently, there have been some research works upon the 316 ASS based on the in-situ EBSD technique, however, most of them focus on the measurement of crystal orientation during the plastic deformation”. Please, cite what works you are talking about. 
  2. Figs. 2 and 4: info in the IPF maps are difficult to read, please enlarge the font size.

Author Response

Investigation of the evolution of Schmid factors (SF) in 316 stainless steel during the in-situ plastic deformation

crystals-2551345

Dear Editor,

Thank you very much for the useful and professional feedback. Please find the revised manuscript. The detailed corrections are listed below point by point:

Reviewer 1:

1. “Currently, there have been some research works upon the 316 ASS based on the in-situ EBSD technique, however, most of them focus on the measurement of crystal orientation during the plastic deformation”. Please, cite what works you are talking about.

Answer: √ Corrected. Please find in revised paper.

  1. Wang X, Liu C, Zhou Z Q , et al. In-situ EBSD investigation of plastic damage in a 316 austenitic stainless steel and its molecular dynamics (MD) simulations. J Mater Res Technol, 2021;13:823-833.
  2. Kamaya M. Measurement of local plastic strain distribution of stainless steel by electron backscatter diffraction.Mater Char, 2009;60:125-132.

2. Figs. 2 and 4: info in the IPF maps are difficult to read, please enlarge the font size.

Answer: √ Corrected. We have made clearly the Fig.2~4. Please find in revised paper.

Reviewer 2 Report

The authors in this paper talked about the Schmid factor which is the geometry factor cos(α)cos(β),  m: τ=σ0⋅m mit m=cos(α)cos(β). The Schmid factor describes the relationship between external normal stresses and the shear stresses caused in a slip system. The article is well-written, however, there are some comments/suggestions for this paper as communication: 

1-there is a lot of typography and grammer errors in the whole manuscript for example: Austenitic stainless steel (ASS), check the capital and small letters of the combined word,  Schmid factor (SF)??? 

2-Line 102: between Loading direction, check the capital letters???

3-this 316 is metastable austenitic or just austenitic????

4-for equipments and software you used add full company and version: ATEX softwares??

5-make introduction a little rich of concept of your research, add someting about stainless steels 316 ASS use this article in your references : https://doi.org/10.3390/met13071268 write some sentences about EBSD and applications of your work.

6-what you mean by this: but less information available on the investigation of the SF change behavior of this material. I cannot understand? re-word it better, the English of the manuscript is difficult to understand making it easy for your readers.

7-EBSD used a lot of time in the whole manuscript but never said its acronym for what words! in the first time you use it, write it completely and then you can use it in the acronym.

8-The quality of figure 1c its awful.......... I cannot even read it.

9-check all of the figure captions for example in figure 1 you have 3 images but in th caption you didn't defined them: Figure 1. In-situ tensile test of the 316 ASS. (a) The specimen coordinates diagram.????

10-Figure 2, what is the mentioned d in the figure??

11- I cannot read some of the figures : figure 1c, 2d, 3a,b,c, SF maps have very low quality make them sharper theya re all pixels!!!

Author Response

Investigation of the evolution of Schmid factors (SF) in 316 stainless steel during the in-situ plastic deformation

crystals-2551345

Dear Editor,

Thank you very much for the useful and professional feedback. Please find the revised manuscript. The detailed corrections are listed below point by point:

Reviewer 2:

1. There is a lot of typography and grammer errors in the whole manuscript for example: Austenitic stainless steel (ASS), check the capital and small letters of the combined word, Schmid factor (SF)?  

Answer: √ Corrected. Please find in revised paper. Besides, the abbreviation SF of Schmid factor is acceptable.

2. Line 102: between Loading direction, check the capital letters?

Answer: √ Corrected. Please find the abstract in revised paper.

3. This 316 is metastable austenitic or just austenitic?

Answer: The material used in this research is just 316 austenitic steel, which do not contain martensitic microstructure.

4. For equipments and software you used add full company and version: ATEX softwares??

Answer: √ Corrected. Please find the abstract in revised paper.

The Zeiss-Sigma 300 type Scanning Electron Microscope equipped with an Oxford-SYMMETRY EBSD detector as well as a Kammrath & Weiss type tensile stage

The version is ATEX-4.03.

5. Makingintroduction a little rich of concept of your research, add someting about stainless steels 316 ASS use this article in your references : https://doi.org/10.3390/met13071268 write some sentences about EBSD and applications of your work.

Answer: √ Corrected. Please find in revised paper.

    5. Rezayat M, Karamimoghadam M, Moradi M, Casalino G, Roa Rovira J J, Mateo A. Overview of Surface Modification Strategies for Improving the Properties of Metastable Austenitic Stainless Steels.Metals, 2023;13(7):1268,1-27. 

6. What you mean by this: but less information available on the investigation of the SF change behavior of this material. I cannot understand? re-word it better, the English of the manuscript is difficult to understand making it easy for your readers.

Answer: √ Corrected. Please find in revised paper.

7. EBSD used a lot of time in the whole manuscript but never said its acronym for what words! in the first time you use it, write it completely and then you can use it in the acronym.

Answer: √ Corrected. Please find in revised paper.

8. The quality of figure 1c its awful.......... I cannot even read it.

Answer: √ Corrected. We have changed the figure 1c.

9. Check all of the figure captions for example in figure 1 you have 3 images but in the caption you didn't defined them: Figure 1. In-situ tensile test of the 316 ASS. (a) The specimen coordinates diagram.

Answer: √ Corrected. We have added them, Please find in revised paper.

10. Figure 2, what is the mentioned d in the figure??

Answer: The relative frequency distribution maps of the Schmid factors of the Origin specimen and the Point-3 specimen.

11.  I cannot read some of the figures : figure 1c, 2d, 3a,b,c, SF maps have very low quality make them sharper theya re all pixels!!!

Answer: √ Corrected. We have make clearly of these figure, Please find in revised paper.
